# Genome-Wide Analysis of MADS-Box Gene Family Reveals *CjSTK* as a Key Regulator of Seed Abortion in *Camellia japonica*

**DOI:** 10.3390/ijms25115770

**Published:** 2024-05-25

**Authors:** Yifan Yu, Xian Chu, Xianjin Ma, Zhikang Hu, Minyan Wang, Jiyuan Li, Hengfu Yin

**Affiliations:** 1State Key Laboratory of Tree Genetics and Breeding, Key Laboratory of Forest Genetics and Breeding, Research Institute of Subtropical Forestry, Chinese Academy of Forestry, Hangzhou 311400, China; yyf95896065@163.com (Y.Y.); chuxian2016@163.com (X.C.); xianjin_ma@163.com (X.M.); huzhikang01@163.com (Z.H.); w524270986@163.com (M.W.); jiyuan_li@126.com (J.L.); 2College of Information Science and Technology, Nanjing Forestry University, Nanjing 210037, China

**Keywords:** *Camellia*, *CjSTK*, seed development, expression analysis, MADS-box gene

## Abstract

The plant MADS-box transcription factor family is a major regulator of plant flower development and reproduction, and the AGAMOUS-LIKE11/SEEDSTICK (AGL11/STK) subfamily plays conserved functions in the seed development of flowering plants. *Camellia japonica* is a world-famous ornamental flower, and its seed kernels are rich in highly valuable fatty acids. Seed abortion has been found to be common in *C. japonica*, but little is known about how it is regulated during seed development. In this study, we performed a genome-wide analysis of the MADS-box gene the in *C. japonica* genome and identified 126 MADS-box genes. Through gene expression profiling in various tissue types, we revealed the C/D-class MADS-box genes were preferentially expressed in seed-related tissues. We identified the AGL11/STK-like gene, *CjSTK*, and showed that it contained a typical STK motif and exclusively expressed during seed development. We found a significant increase in the *CjSTK* expression level in aborted seeds compared with normally developing seeds. Furthermore, overexpression of *CjSTK* in *Arabidopsis thaliana* caused shorter pods and smaller seeds. Taken together, we concluded that the fine regulation of the *CjSTK* expression at different stages of seed development is critical for ovule formation and seed abortion in *C. japonica*. The present study provides evidence revealing the regulation of seed development in *Camellia*.

## 1. Introduction

The plant MADS-box gene family is an important regulator of plant growth and development [1,2]. The MADS-box is named after the first letters of the transcription factors that were first discovered: *MINICHROMOSOME MAINTENANCE1* (*MCM1*) in *Saccharomyces cerevisiae* [3], *AGAMOUS* (*AG*) in *Arabidopsis thaliana* [4], *DEFICENS* (*DEF*) in *Antirrhinum majus* [5] and the *serum response factor* (*SRF*) in *Homo sapiens* [6]. In plants, MADS-box genes are involved in the control of multiple developmental processes, including flowering time, maintenance of meristem activity, determination of floral organ properties, and fruit development and maturation [2,7,8]. With the emergence of a large number of studies, many insights into MADS-box regulatory functions have been achieved at the molecular mechanistic level, and the diversity of their functions has been gradually revealed [9,10,11].

MADS-box proteins contain a highly conserved DNA-binding domain of approximately 50–60 amino acids in length in their N-terminal region [1], and this domain could be involved in recognizing and binding the CArG motif of their target gene [12]. Based on gene and protein structures and evolutionary relationships, the MADS-box gene family can be divided into two major types: type I and type II (MIKC type). The type I MADS-box genes can be further classified into Mα, Mβ, Mγ, Mδ subclasses [13], with preferential interactions between subfamilies forming heterodimeric complexes [14,15,16]. Type II MADS-box genes have a special MIKC structure, which is composed of an N-terminal MADS domain, the I (intervening), K (keratin-like) regions and a variable C-terminal transcriptional activation domain [17,18]. During flower development, the MADS-box gene is a crucial factor that regulates the identity and development of floral organs [19,20]. According to the classical ABC model, MADS-box genes of different functional categories (e.g., A-function, B-function, C-function, D-function, etc.) form a complex regulatory network through interactions to determine cell differentiation [19,21]. Seed development was found to be regulated by C/D-like MADS-box genes [2,22]. Among them, *SEEDSTICK/AGAMOUS-LIKE 11* (AGL11/STK) is a key class of C/D-class MADS-box genes that regulates ovule development [22]. In *Arabidopsis*, *STK*, as well as *SHATTERPROOF1* (*SHP1*) and *SHATTERPROOF2* (*SHP2*), are required for ovule development and seed dehiscence [20,23]. *STK*, *SHP1* and *SHP2* have redundant regulatory roles in ovule development [23], and mutations in *STK* have prevented normal ovule formation and seed development, which demonstrates the importance of *STK* in ovule development [23]. In addition, *STK* is also involved in the development of ovule stalks and fruit abscission [23].

In various plants, AGL11/STK-like genes have been found to have conserved functions and are extensively involved in the regulation of ovule development in plants. In petunias, *FLORAL BINDING PROTEIN 7* (*FBP7*) and *FLORAL BINDING PROTEIN 11* (*FBP11*) are homologous genes of the AG11/STK clade, and the inhibition of their normal expression results in the absence of ovule production in the placenta [24,25,26]. In rice, *OsMADS13*, a homolog of *STK*, was found to determine ovule formation, and in the *osmads13* mutant, ovules failed to form and were transformed into a carpel-like structure [27]. In grapevines, *VviAGL11* promotes lignification of the endosperm, a process that is a prerequisite for endosperm growth and embryo development; and disrupting *VviAGL11* expression by RNAi in seeded grapevine varieties results in a significant reduction in both seed size and number [28,29]. In oil palm, mutations in the *SHELL* gene (the *STK* homolog), which controls ovule characteristics and seed development, causes the loss of the encapsulated cuticle of the seed kernels [30].

AGL11/STK-like genes can directly or indirectly regulate the expression of downstream genes involved in multiple biological pathways. In *Arabidopsis thaliana*, a transcriptomic analysis of developing ovules and seeds showed that *STK* prevents proanthocyanidin (PA) accumulation and controls the development and differentiation of endosperm by repressing *BANYULS*/*ANTHOCYANIDIN REDUCTASE* (*BAN*/*ANR*) expression [31]. In the cytokinin (CK) hormone pathway of fruit development, *STK* was able to directly bind to the promoter of *Cytokinin oxidase*/*dehydrogenase* (*CKX7*) and regulated the degradation of CK, which in turn promoted fruit growth [32]. During seed pod development, *STK* can indirectly regulate *FRUITFUL* (*FUL*) expression and modulate the associated developmental pathways [33]. In *Arabidopsis*, *STK* is tightly restricted to be expressed during seed development, a process that is regulated by epigenetic modification pathways [34,35]. It was found that BASIC PENTACYSTEINE (BPCs) transcription factors and *SHORT VEGETATIVE PHASE* (*SVP*) were essential regulators for the suppression of *STK* expression [32]. In the promoter of *STK*, *BPCs* and *SVP* maintained their level of H3K27me3 through recruitment of *LIKE HETEROCHROMATIN PROTEIN1* (*LHP1*), which was required for inhibiting *STK* expression in vegetative organs [36]. In addition, *AGL11*/*STK-like* genes were found to play regulatory roles in other biological processes. Studies on the *BnaAGL11* gene, an *STK* homologue in oilseed rape, showed that *BnaAGL11* regulated leaf morphogenesis and promoted leaf senescence [37]. Overexpression of *SlAGL11* in tomatoes resulted in increased fruit softness and sugar content [34]. Studies on several dioecious plants showed that *STK* is one of the key genes for female flower development, which might be closely related to its differential expression levels in different floral organs [38,39,40,41].

*Camellia japonica* L. is a traditional ornamental flower across the world, with variable flower shapes and a variety of flower colors. *C. japonica* seeds are rich in oil, which can be utilized in high value products (e.g., cosmetics and food). *C. japonica* fruits usually contain 2–3 chambers, with about 3–4 fertilized ovules per chamber during the early stage of fruit development, but only 1–2 mature seeds can develop per chamber. Currently, there are very few studies on the molecular regulation of ovule formation in *C. japonica*, and the understanding of seed abortion remains unclear. In this study, we identified the MADS-box genes at the genome-wide level in *C. japonica* and revealed potential regulatory factors involved in flower, fruit and seed development based on gene expression and sequence analysis. We focused on the *AGL11*/*STK* homologous gene, *CjSTK*, and investigated the regulatory role of *CjSTK* in ovule formation and seed abortion through gene expression and functional studies. Our study suggests that the regulation of *CjSTK* expression at different developmental stages during seed formation is the key to ovule formation and seed abortion.

## 2. Results

### 2.1. Genome-Wide Identification and Characterization of MADS-Box Genes in C. japonica (cjND) Genome

We performed MADS-box gene family identification based on sequence and conserved motif information in *C. japonica*. In total, we identified 126 MADS-box genes from the *C. japonica* (cjND) genome [42]. We found that the gene lengths of 53% of these genes were below 3 kb, and 13 genes were longer than 20 kb (Figure 1A). The number of exons in genes varies widely, with about 48% of genes containing only one or two exons and about 42% of genes having no less than six exons (Figure 1B). The predicted Open Reading Frame ranged from 243 bp to 1827 bp, with more than half of the genes having amino acid numbers concentrated in the 200–300 region (Figure 1C).

We analyzed the chromosomal localization information of *C. japonica* MADS-box genes and found that 120 members were localized in the 15 assembled chromosomes and six MADS-box genes (EVM0019351, EVM0000454, EVM0020439, EVM0008457, EVM0008655, EVM0020882) were found to be localized in unassembled contigs (Figure 1D). We showed that chromosome 6 had the highest number, containing 22 MADS-box genes (Figure 1D). In consideration of gene family expansion, we showed there were 13 tandem clusters of MADS-box genes, including 36 genes. These results provide fundamental resources for the further analysis of MADS-box genes in *Camellia* plants.

### 2.2. Phylogenetic Analysis of MADS-Box Genes in C. japonica

To further analyze the evolutionary relationships, we classified the MADS-box genes according to the functional information of *Arabidopsis* [43]. The results showed that 126 MADS-box genes were categorized into type I and type II, of which 65 type I genes contained Mα (31), Mβ (15) and Mγ (19), and 61 type II genes contained MIKC (47) and Mδ (14) types (Figure 2A). MIKC-type MADS-box genes have been found to be critical regulators in determining the identity of floral organs [18], and the canonical ABCE model genes have been found to be widely conserved in plants [19,21]. Through phylogenetic analysis, we identified homologs of class A-, B-, C/D- and E-classes of floral regulators in *C. japonica*, which provides a basis for future functional analysis.

To investigate the potential candidates involved in the floral development, we investigated the gene expression profiles containing 14 tissue types from leaves, flower fruits and seeds of *C. japonica* [44,45]. We identified 62 MIKC-type MADS-box genes with relatively high expression levels in the examined samples [fragment per kilobase of transcript per million fragments (fpkm) > 10 in all 42 samples], suggesting those are key regulators involved in the reproductive development. Further, we found that the expression profiles of these genes had distinctive expression patterns in different tissue types (Figure 2B). We focused on the genes involved in carpel and seed development and found that a subgroup of C/D-class genes were highly expressed in fruit and seed tissues (Figure 2B). Particularly, the ortholog of *STK* (EVM0019646.1, renamed as *CjSTK*) was predominantly in embryo tissues and identified as a key regulator of seed development (Figure 2B).

### 2.3. Identification and Characterization of AGL11/STK Homolog in C. japonica

To investigate the potential function of the *STK* ortholog in *C. japonica*, we performed 5′ and 3′ *RACE* analysis to obtain the full coding sequences of *CjSTK*. The full-length transcript of *CjSTK* was identified, encoding a 696 bp CDS identical to EVM0019646.1 (NCBI GenBank accession UYB00957). To investigate the phylogenetic relationships, we used some relevant C/D-class genes, as well as AGL11/STK genes from different species to construct evolutionary trees. We found that there were three major clades of C/D-class genes, including AG clade, PLE clade and AGL11/STK clade (Figure 3A); and the *CjSTK* (EVM0019646.1) was identified as the ortholog of *STK* in *C. japonica* (Figure 3A). To further evaluate the conservation of AGL11/STK genes, we performed a sequence alignment of members of the AGL11/STK clade from different plants. We showed that the AGL11/STK clade had conserved MADS-box structures, and a C-terminal motif (termed as STK motif) was revealed (Figure 3B). Through the protein structure simulation, the N- and C-terminus were revealed to be flexible regions potentially involved in protein–DNA interactions (Figure 3C).

To investigate the expression profile of *CjSTK* in *Camellia*, we measured *CjSTK* transcript levels in various organs, including sepal, petal, stamen, carpel, exocarp, fruit, seed coat and seed kernel (Figure 3D). We showed that the *CjSTK* transcripts were only detected in fruit and seed tissues, with the highest expression level in seed kernels (Figure 3D). The expression pattern supports *CjSTK* being a regulator of seed development.

The fruit of *C. japonica* usually has three seed chambers and at the early stage of development, there are about three fertilized ovules per chamber (Figure 4A–D); seed abortion is commonly seen at the late stage of development (Figure 4B–D). In order to explore the potential role of *CjSTK*, we analyzed the development of *Camellia* seeds and divided them into normal and aborted types according to their relative size (Figure 4E). We showed that the relative expression of *CjSTK* in two types of seed kernels varied and the expression level of *CjSTK* in seed kernels at the early stage of abortion was significantly higher (about two-fold upregulated) than that in seeds with normal development (Figure 4E,F). Taking these results together, we propose that the regulation of the expression level of *CjSTK* might be critical for seed development and seed abortion.

### 2.4. Ectopic Expression of CjSTK in Arabidopsis Causes Defects in Seed and Silique Development

To investigate the role of *CjSTK*, we determined the subcellular localization of *CjSTK* by using a transient expression assay of a *CjSTK-GFP* fusion protein in tobacco (*Nicotiana benthamiana*). We found that the *CjSTK-GFP* fusion protein was co-located with the 35S:H2B:mCherry nuclear localization marker, indicating *CjSTK* is predominantly localized to the cell nucleus (Figure 5).

To study the functions of *CjSTK*, we obtained the over-expression lines of *CjSTK* in *Arabidopsis*. The construct-specific amplification of transgenic lines was performed to identify positive transgenic lines (Figure 6A). Furthermore, we examined the expression levels of *CjSTK* by the qRT-PCR analysis in T2 positive transgenic lines (Figure 6B). We showed that the expression of *CjSTK* was detected in all of the overexpression lines but not in the WT (Figure 6B), indicating an ectopic expression of *CjSTK*.

We showed that the transgenic lines had smaller and deformed siliques compared to wild-type *Arabidopsis* (Figure 7A). We analyzed the silique length of wild-type, *CjSTK-OX1* and *CjSTK-OX3* lines and showed that the silique length of the transgenic strain was significantly shorter than that of the wild-type (Figure 7B). We also observed the transgenic lines had smaller seeds compared to that of wild-type (Figure 7C). We showed that the seed length, seed width and seed area of the transgenic lines were significantly smaller than those of the wild-type (Figure 7D–F). These results indicate that *CjSTK* plays an import role in seed development and the constitutive expression of *CjSTK* can inhibit seed growth.

## 3. Discussion

### 3.1. Conservation and Evolution of the MADS-Box Gene Family in Camellia japonica

Based on the reference genome of *C. japonica* [42], we systematically characterized the MADS-box family at the genome-wide level (Figure 1). We found that the classification and composition of the MADS-box family in *C. japonica* are similar to other closely related plants [47,48,49], which implies that they may also have conserved functions in multiple aspects of growth and development. Through the functional information and phylogenetic relationships, we identified some homologous genes, such as A-, B-, C- and D-class genes of flower development (Figure 2A). We used previous transcriptomes of floral organs, fruits and other tissues in *C. japonica* to reconstruct a set of gene expression profiles, including 14 tissue types using the reference genome information (Figure 2B). Our results indicated that members of the MADS-box family have distinct expression patterns in different tissue types (Figure 2B). Therefore, using this data in combination with phylogenetic analysis, we can uncover important regulatory genes from the pattern of gene expression. Using synteny analysis of the *C. japonica* genome, previous studies have identified large segment duplications of chromosome chr8, leading to two paralogs of the C-class genes (*AG* homologs) [42]. In the current study, our expression patterns and phylogenetic results are consistent with previous results (Figure 3A), supporting the conserved functions of the C-class genes in the regulation of floral development. We identified a homolog of a class D gene, and through sequence analysis and expression profiling studies (Figure 2 and Figure 3), we concluded that *CjSTK* is an ortholog of *Arabidopsis STK* in *Camellia*. Taken together, the genome-wide identification of MADS-box provides a basis for subsequent studies on the functions of MADS-box genes in the molecular regulation of the development of flowers, fruits and seeds.

### 3.2. Dual Function of the AGL11/STK Genes in Regulating Seed Development

According to the conventional ABCE model, the D function mainly refers to the property determination of the ovule and seed formation, and the loss of function of class D genes usually results in the failure of seed formation [29,32,33]. However, loss-of-function mutants often eliminate the regulatory roles at later stages of seed development. In addition, in some plants, the function of class D genes is involved in other aspects of seed formation. For example, in both grape and oil palm, studies have found that the homologous gene of *STK* can promote the lignification of the seed coat [32,50]. These results suggest that class D genes may have multiple functions. In *C. japonica*, we suggest that *CjSTK* has a dual function, participating in the establishment of ovule properties and their formation at early stages of development and regulating specialized cell formation and the growth of seeds at later stages. Our evidence is mainly that *CjSTK* is the only class D gene identified from the *C. japonica* genome in the present study (Figure 2), and its expression is specific to seed-associated tissues (Figure 2B and Figure 3B). These results suggest that *CjSTK* is an important regulator of seed development, probably sharing the conserved class D function in *C. japonica*. In late seed development, we found that the expression level of *CjSTK* was significantly higher in aborted seeds than in normal seeds (Figure 4). This suggests that the fine regulation of *CjSTK* can be involved in the differentiation of specific tissues in seeds. The constitutive expression of *CjSTK* in *Arabidopsis* mainly resulted in a phenotype of growth inhibition in pod and seed development (Figure 7). Thus, in light of the previous analysis, we suggested that the dual role of *CjSTK* is involved in the fine expression of *CjSTK* at the later stages of seed development.

### 3.3. Activation and Repression of CjSTK Gene Expression Is Critical for Maintaining Seed Development

The expression of STK/AGL11 genes is regulated by epigenetic modification. In *Arabidopsis*, class I and II BPC redundantly suppressed *STK* expression by recruiting transcriptional inhibitory complexes and maintaining the level of H3K27me3 [35,36]. DNA methylation at the promoter region was found to be a key factor controlling the transcription level of *CmAGL11* in chestnuts during somatic embryogenesis [51]. In *Camellia*, it has been proven that the class I BPC transcription factor CjBPC1 can directly bind to the promoter sequences of *CjSTK* to inhibit its expression [38]. Therefore, the BPC-mediated regulation of *CjSTK* might be required for the development of ovules in *C. japonica*. Previous studies on cotton found that the overexpression of *GhMADS14*, a homologous gene of STK/AGL11, caused a decrease in endogenous GA levels, leading to a smaller size of seeds in Arabidopsis [52]. In Arabidopsis thaliana, it has been found that *STK* can directly bind the promoter of *CKX7* to control the degradation of CK [32]. Thus, ovule and seed development defects in *CjSTK* transgenic plants may be associated with altered hormonal pathways, and to obtain further functional understanding requires analysis of the downstream genes of *CjSTK*. Although we propose that the late function of *CjSTK* involves repression of expression, this does not indicate that *CjSTK* can be missed for seed development because in the aborted seed, the *CjSTK* maintained the expression (Figure 4). Thus, it can be inferred that the function of *CjSTK* in regulating seed development is the result of the fine regulation of its transcriptional activation and transcriptional repression. We hypothesize that the regulation of seed development by *CjSTK* may involve multiple pathways, such as hormone signaling and epigenetic pathways, which need to be further explored.

## 4. Materials and Methods

### 4.1. Plant Materials and Growth Conditions

*Arabidopsis thaliana* Ecotype Columbia (Col-0) was used in this study. *C. japonica* ‘Naidong’ was obtained from the camellia nursery of the Research Institute of Subtropical Forestry, Chinese Academy of Forestry (119°57′ N, 30°04′ E; Fuyang, Zhejiang, China), under natural growth conditions. For wild-type *Arabidopsis*, the seeds were sterilized and sown on a solid medium containing 1/2 Murashige and Skoog (MS) medium, and for transgenic *Arabidopsis*, 10 mg/L of hygromycin was added into the medium for selection. When the seedlings reached the two-cotyledon stage, they were transferred to plastic pots filled with a nursery substrate (peat:vermiculite:perlite = 2:1:1) and placed in a greenhouse (long-day conditions, 16 h light/8 h dark), set at 22 °C.

### 4.2. Gene Family Analysis and Expression Profiling of MADS-Box Genes

The cjaND genome and gene annotation were used for gene identification [42]. The HMMER3.1b2 version (February 2015) [53] was used to identify the MADS-box gene family through both the K-domain (PF01486) and the MADS-domain (PF00319) using default parameters [54]. We found both PFAM domains yielded the same 126 high-confidence candidates, which were used for further analyses. The RNA-sequencing datasets from floral and fruit organs were obtained based on previous studies [44,45]. The gene expression analysis was performed based on the Hisat2-Stringtie pipeline as described [55].

### 4.3. Isolation of Total RNA and Real-Time PCR Analysis

To isolate the total RNA, an EASYspin Plus Plant RNA Kit (Aidlab, Beijing, China) was used as per the manufacturer’s instructions. To remove DNA, RNase-free DNase treatment was implemented with a TaKaRa kit (Dalian, China). About 1 μg of RNA was then utilized for reverse transcription. The iScript cDNA Synthesis Kit (TaKaRa, Dalian, China) was utilized for the first-stand cDNA synthesis. Real-time PCR was conducted using a SYBR^®^Premix Ex Taq™ (TaKaRa, Dalian, China) kit in accordance with the manufacturer’s instructions. To normalize the gene expression data using the relative quantification method (2^−ΔΔCT^) [56], *ACTIN* and *GAPDH* were the chosen as internal reference genes for *Arabidopsis* and *Camellia*, respectively. The experiment was repeated three times with two biological replicates. The primers for expression analysis are listed in Table 1.

### 4.4. Gene Cloning, Vector Construction and Subcellular Localization Analysis

To obtain *CjSTK* genes, 5′ and 3′*RACE* (rapid amplification of cDNA ends) experiments were performed to amplify the full transcript by using a SMARTer *RACE* kit (Clontech, Palo Alto, CA, USA) using gene-specific primers (Table 1). For vector construction, the full length sequences of *CjSTK* gene were cloned, and the CDS without a stop codon was cloned into a pEXT06/g vector (Baige, Suzhou, China) to generate an *CjSTK-GFP* fusion construct driven by the 35S promoter. The leaves of *N. benthamiana* were transformed by *Agrobacterium* (pGV3101) (WEIDI, Shanghai, China) and then were dark-cultured for 2–3 days. Florescence signals were captured by confocal laser scanning microscopy (LSM900, Zeiss, Oberkochen, Germany).

### 4.5. Arabidopsis Transformation and Screening of CjSTK-OX Transgenic Lines

The plasmid of 35S::*CjSTK* was transformed into *Agrobacterium tumefaciens* strain C58 (pGV3101), and then used for the transformation of *A. thaliana* by the floral-dip method [57]. The transformed Arabidopsis T0 seeds were sown on a 1/2 MS solid medium with 10 mg/L of hygromycin for selection. The genomic DNA of transgenic plants and WT was extracted from the rosette leaves (two weeks old) using the CTAB method, and PCR validation was performed using hygromycin-specific primers to determine positive transgenic lines.

### 4.6. Statistical Analysis

The Student’s *t*-test analyses were used for to determine the statistical significance by R Core Team (2023) and GraphPad Prism 9.3 Software (Boston, MA, USA, www.graphpad.com (accessed on 16 June 2022)).

### 4.7. Phylogenetic Tree Construction

MADS-box gene family sequences from *Arabidopsis* were obtained from PLAZA3.0 [58], and the sequence alignment and initial neighbor-joint tree was constructed using TBtools [59]. The ITOL-tree online tool was used to edit and polish the original tree file for presentation [60]. For the AGL11/STK phylogenetic analysis, MEGA7.0 was used for alignment and tree construction [61]. The JTT matrix-based method with a gamma distribution (shape parameter = 0.931) was used.

### 4.8. Histological Analysis

To observe the morphological changes of seed abortion, the young fruit of *C. japonica* ‘Naidong’ was fixed by using a FAA (50% ethanol, 5% acetic acid, 10% formalin) solution. The fruits with diameters from 3 to 7 mm were used for sections preparation and staining and microscopy analyses as described [62].

## Figures and Tables

**Figure 1 ijms-25-05770-f001:**
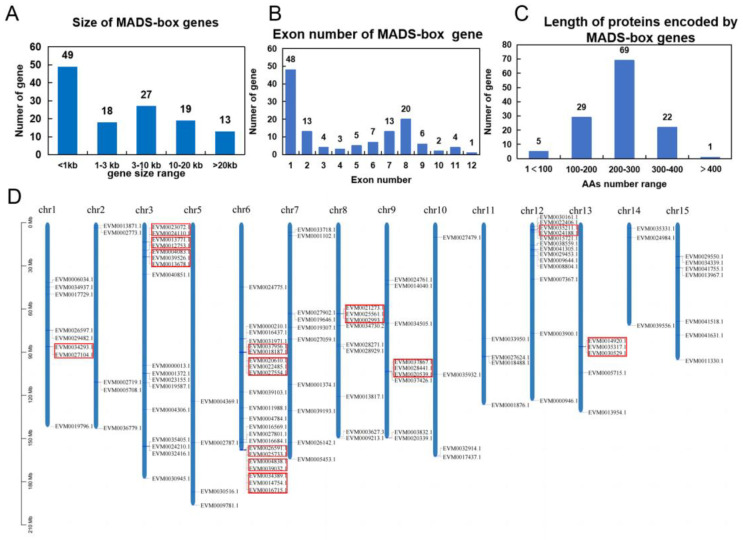
The overview of MADS-box genes in *C. japonica* cjaND genome. (**A**) the size distribution pattern of the lengths of the MADS-box genes identified; (**B**) the number of exons in MADS-box genes; (**C**) the length of proteins potentially encoded by MADS-box genes; (**D**) the distribution of MADS-box genes in chromosomes of the cjaND reference genome. The red rectangles indicate the tandem clusters of MADS-box genes.

**Figure 2 ijms-25-05770-f002:**
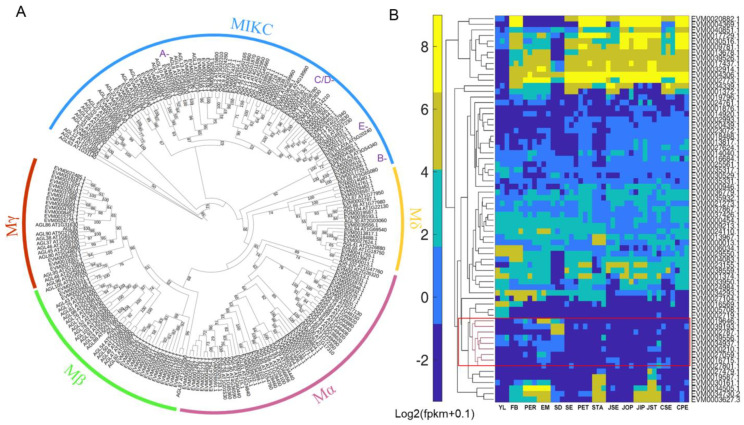
Characterization and expression profiling of MADS-box genes in *C. japonica*. (**A**) The phylogenetic tree of 126 MADS-box genes from cjaND genome and 101 MADS-box genes from *Arabidopsis thaliana*. The colored arc lines indicate the subgroups of MADS-box gene family; the colored letters indicate the canonical ABCE model genes. (**B**) The heatmap expression of 42 samples of 14 tissues types from previous RNA-seq studies in *C. japonica*. The 62 MIKC-type MADS-box genes are revealed. The red rectangle indicates the genes with specific expression in seed-related tissues. YL, young leaf; FB, floral buds; PER, pericarp; EM, endocarp; SD, seed kernel; SE, sepal; PET, petal; STA, stamen; JSE, sepal in ‘Jinpanlizhi’; JOP, outer petal in ‘Jinpanlizhi’; JIP, inner petal in ‘Jinpanlizhi’; CSE, sepal in ‘Chidan’; CPE, petal in ‘Chidan’.

**Figure 3 ijms-25-05770-f003:**
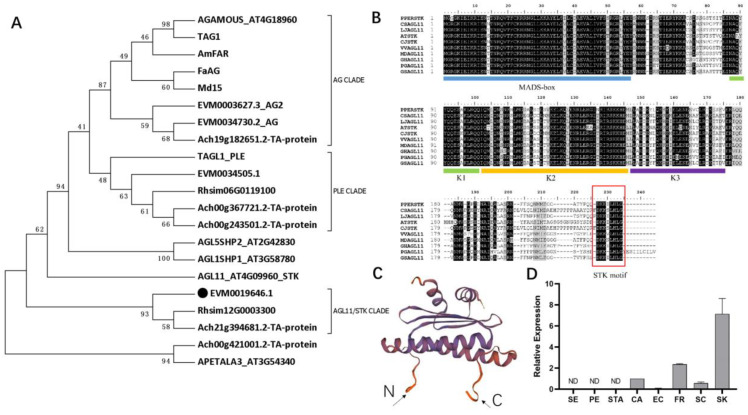
*CjSTK* is an ortholog of AGL11/STK in *C. japonica*. (**A**) Phylogenetic relationships of C/D-class genes from various plants. The details of the sequences are provided in the Appendix A. (**B**) Protein sequence alignment of *AGL11/STK* genes. The C-terminal STK motif is indicated by the red rectangle. (**C**) The 3D structure model of *CjSTK* created by homology modeling [46], and the arrows indicate the N- and C-terminus. (**D**) Real-time PCR analysis of *CjSTK* expression in various tissues of *C. japonica*. ND, not detected. SE: sepal; PE: petal; STA: stamen; CA: carpel; EC: exocarp; FR: fruit; SC: seed coat; SK: seed kernel.

**Figure 4 ijms-25-05770-f004:**
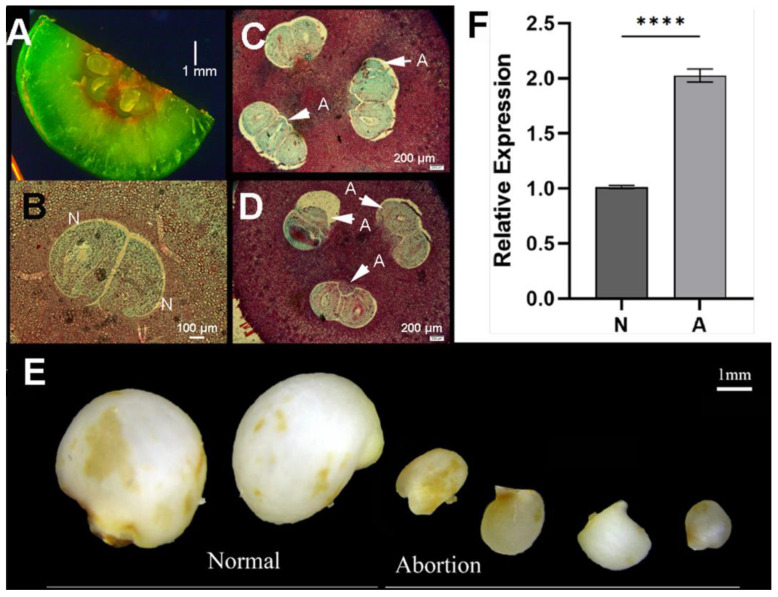
*CjSTK* is highly expressed in aborted seeds at a late stage of development. (**A**) Anatomy of a dissected fruit in *C. japonica* and multiple (2–4) ovules can be seen in the seed chamber. (**B**–**D**) The cross-section of incipient fruits in *C. japonica* containing normal and aborted seeds. N, normal seeds; A, aborted seeds. (**E**) The late stage of normal and aborted seeds. (**F**) The relative expression of *CjSTK* in normal and aborted seeds. Stars indicate significant changes by a Student’s *t*-test at **** *p* < 0.0001.

**Figure 5 ijms-25-05770-f005:**
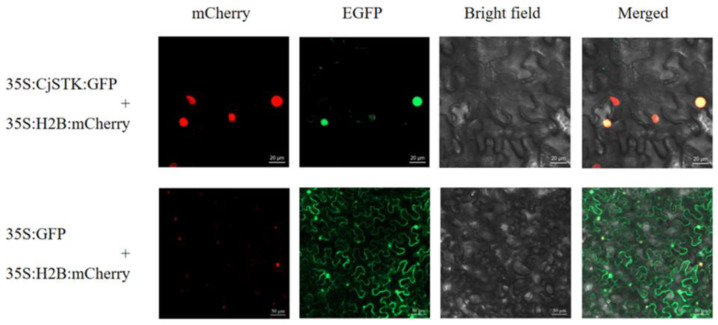
Subcellular localization of *CjSTK*. An agro-infiltration of the tobacco leaf of *CjSTK-GFP* was performed. The H2B::mCherry (red) nuclear localization was co-infiltrated, and the free *GFP* (green) was used as a control. Bar = 200 µm.

**Figure 6 ijms-25-05770-f006:**
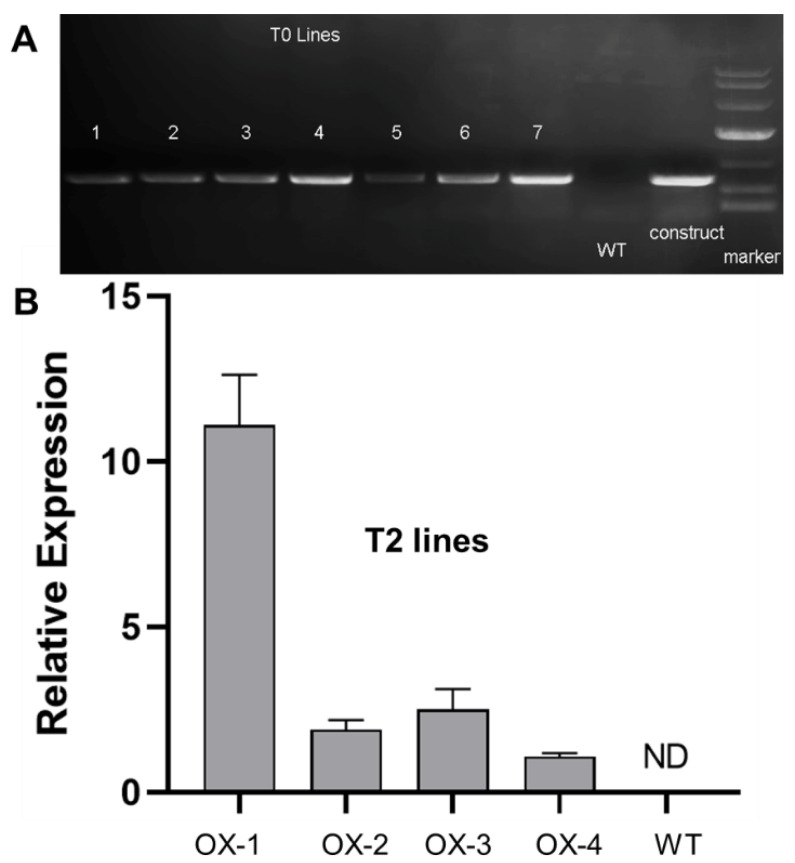
Validation of *35S*::*CjSTK* transgenic lines in *Arabidopsis*. (**A**) The construct-specific amplification of T0 transgenic lines. WT, wild-type plant; construct, the positive control of the used vector. (**B**) Expression of *CjSTK* in the leaves of WT and *CjSTK-OX* transgenic plants, determined by qRT-PCR. ND, not detected. Error bars indicate standard deviations of all the replicates in each genetic background.

**Figure 7 ijms-25-05770-f007:**
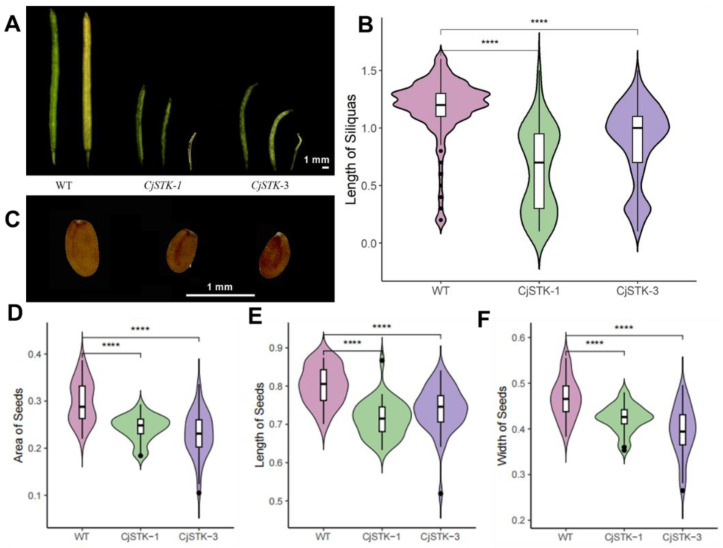
Phenotypical characterization of transgenic *CjSTK* lines in *Arabidopsis*. (**A**) The morphologies of typical siliques in wild-type and overexpressed *CjSTK* lines, scale bar = 1 mm. (**B**) The statistical analysis of the pod lengths of wild-type and two independent transgenic lines. (**C**) Phenotypes of mature seeds in wild-type and transgenic *CjSTK* lines, scale bar = 1 mm. (**D**–**F**) The statistical analysis of seed area (**D**), seed length (**E**) and seed width (**F**) of wild-type and two independent transgenic lines. The stars indicate significant changes (**** *p* < 0.0001) by Student’s *t*-test analysis.

**Table 1 ijms-25-05770-t001:** Primer sequences used in the study.

Primer Name	Primer Sequence (5′→3′)	Use
*CjSTK-F*	ATGGGGCGAGGAAAAATTG	Gene cloning
*CjSTK-R*	TTATCCAAGATGGAGAGAC	
*ExCjSTK-F*	ATGGGGCGAGGAAAAATTG	Vector construction
*ExCjSTK-* *R*	TTATCCAAGATGGAGAGAC	
*GAPDH-F*	GGGAATCCTTGGTTACACTGAG	Internal control in *C. japonica*
*GAPDH-R*	ACCCCATTCGTTGTCATACC	
*qCjSTK-F*	CATTCCTTCGAGCCAAGATAGC	qRT-PCR for *CjSTK*
*qCjSTK-R*	TCTCCTGGCACCATGTTTTG	
*AtACTIN-F*	GCACCCTGTTCTTCTTACCG	Internal control in *Arabidopsis*
*AtACTIN-R*	AACCCTCGTAGATTGGCACA	
*HYG_F*	TGACCTATTGCATCTCCCGC	Hygomycin selection
*HYG_R*	ATTTGTGTACGCCCGACAGT	
5′*RACE-F*	GCTGTCAACGATACGCTACGTAAC	5′*RACE* for *CjSTK*
5′*RACE-R*	CTCATAGACGCGTCCTCGACTA	Gene-specific primer
3′*RACE-F*	AGTCACATTCTGCAAGCGGAGAA	3′*RACE* for *CjSTK*
3′*RACE-R*	GCGAGCACAGAATTAATACGACT	Gene-specific primer

## Data Availability

All data in this manuscript are available on the associated website.

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
