# Peer review of "Genome-Wide Analysis of MADS-Box Gene Family Reveals CjSTK as a Key Regulator of Seed Abortion in Camellia japonica"

_ijms, 2024, doi:10.3390/ijms25115770_

Round 1
Reviewer 1 Report
Comments and Suggestions for Authors
Authors in the present manuscript have analysed MADS-box gene family in Camellia japonica where they have identified 126 MADS-box genes. Further, based on the expression pattern in various seed related tissues, they identified CjSTK gene is exclusively expressed during seed development. Its function has also been confirmed by overexpression in Arabidopsis thaliana. The experiments are well planned and systematic. I have few suggestions
Introduction section is little lengthy, it can be improved by making it short by keeping only most relevant information.
The authors are proposing the relation between CjSTK and hormones like GA and cytokinin based on previous reports in other crops. To prove this in Camelia, the authors can correlate the expression of CjSTK with hormone levels by simply measuring the hormones in tissues.
English language in material and methods need to be improved for grammer.
Some minor points are..
Line 52: replace identify with identity
Subheading 2.3: Figures, tables and schemes might be misplaced here. please check the heading again and correct it.
Correct the numbering in subheadings. After 2.3 there is 2.5.
Line247: delete the word ‘seeds’ and correct sentence as ‘we also observed the transgenic lines had…’
Line 319: merge the two sentences as ‘…..plants may be associated with altered hormonal pathways, and to obtain further functional understanding….’
Line 354: ‘….the relative quantification method (2-ΔΔCT) [59], ACTIN and GAPDH were chosen as internal….’
Comments on the Quality of English Language
Language of material and methods need to be improved
Author Response
Authors in the present manuscript have analysed MADS-box gene family in Camellia japonica where they have identified 126 MADS-box genes. Further, based on the expression pattern in various seed related tissues, they identified CjSTK gene is exclusively expressed during seed development. Its function has also been confirmed by overexpression in Arabidopsis thaliana. The experiments are well planned and systematic. I have few suggestions
#Thank you for your comments. We have addressed all of your concerns in this revised manuscript. Please find details below.
Introduction section is little lengthy, it can be improved by making it short by keeping only most relevant information.
#In this revision, we have shortened the introduction. Please find the revised version of Introduction for details. Thank you
The authors are proposing the relation between CjSTK and hormones like GA and cytokinin based on previous reports in other crops. To prove this in Camelia, the authors can correlate the expression of CjSTK with hormone levels by simply measuring the hormones in tissues.
#Thank you. The comment is sensible. This study focuses on the regulation of CjSTK in relation to seed development. Your comment on the hormone levels is very meaningful, and we believe that the function of CjSTK is closely related to hormones. In the follow-up study, we will focus on both the potential downstream genes and the interactions proteins of CjSTK, in order to clarify the regulatory roles of phytohormones.
English language in material and methods need to be improved for grammer.
#In this revised manuscript. We have paid attention to the writing of the manuscript, and particularly on the section of material and methods.
Some minor points are..
Line 52: replace identify with identity
#RESPONSE: We are sorry for the misleading. We have changed the word in the revised manuscript:
Line 51-52: During flower development, the MADS-box gene is a crucial factor that regulates the identity and development of floral organs [19, 20].
Subheading 2.3: Figures, tables and schemes might be misplaced here. please check the heading again and correct it.
#RESPONSE: We are sorry for the oversight. We have corrected the heading:
2.3. Identification and characterization of AGL11/STK homolog in C. japonica
Correct the numbering in subheadings. After 2.3 there is 2.5.
#RESPONSE: Thank you for your careful reading. We have corrected the numbering:
2.4. Ectopic expression of CjSTK in Arabidopsis causes defects of seed and silique development
Line247: delete the word ‘seeds’ and correct sentence as ‘we also observed the transgenic lines had…’
#RESPONSE: We are sorry for the oversight. We corrected the wrong grammar and corrected the sentence in the revised manuscript:
Line247-248: We also observed the transgenic lines had smaller seeds compared to that of wild type (Fig. 7C).
Line 319: merge the two sentences as ‘…..plants may be associated with altered hormonal pathways, and to obtain further functional understanding….’
#RESPONSE: Thank you for the comment. We have merged the two sentences:
Line317-320: Thus, ovule and seed development defects in CjSTK transgenic plants may be associated with altered hormonal pathways, and to obtain further functional under-standing requires analysis of the downstream genes of CjSTK.
Line 354: ‘….the relative quantification method (2-ΔΔCT) [59], ACTIN and GAPDH were chosen as internal….’
#RESPONSE: Thank you for the comment. We have merged the two sentences:
Line354-356: To normalize the gene expression data using the relative quantification method (2-ΔΔCT) [59], ACTIN and GAPDH were the chosen internal reference genes for Arabidopsis and Camellia, respectively.
Reviewer 2 Report
Comments and Suggestions for Authors
In this manuscript, Yu and co-authors present an interesting study that highlights the involvement of CjSTK as a key regulator of seed abortion in the plant Camellia japonica, following a genome-wide analysis of MADS-box gene family of proteins. The authors describe a significant increase of CjSTK expression in aborted seeds compared with normally developing seeds, while heterologous overexpression of CjSTK in Arabidopsis thaliana resulted in shorter pods and smaller seeds. The study is timely, being in line with studies regarding regulation of plant growth and development. The manuscript is coherent and logical, presenting sound and solid data in a comprehensive way.
In this reviewer’s eyes, there are only minor (especially editing) issues that the authors need to address before the manuscript can be accepted for publication, some of which are listed below.
- Figure 1, A: Please re-formulate the phrase “gene model”. Maybe “genes identified?”
- Figure 1, C: Please re-formulate “AAs number of MADS-box genes”. For example, with “the length of proteins potentially encoded by MADS-box genes”.
- Figure 2, legend: please italicize “Arabidopsis thaliana”.
- Line 111: “C. japonica” has to be italicized.
Author Response
In this manuscript, Yu and co-authors present an interesting study that highlights the involvement of CjSTK as a key regulator of seed abortion in the plant Camellia japonica, following a genome-wide analysis of MADS-box gene family of proteins. The authors describe a significant increase of CjSTK expression in aborted seeds compared with normally developing seeds, while heterologous overexpression of CjSTK in Arabidopsis thaliana resulted in shorter pods and smaller seeds. The study is timely, being in line with studies regarding regulation of plant growth and development. The manuscript is coherent and logical, presenting sound and solid data in a comprehensive way.
In this reviewer’s eyes, there are only minor (especially editing) issues that the authors need to address before the manuscript can be accepted for publication, some of which are listed below.
#Thank you for your comments. We have addressed all of your concerns in this revised manuscript. Please find details below.
- Figure 1, A: Please re-formulate the phrase “gene model”. Maybe “genes identified?”
#RESPONSE: We are sorry for the misleading. We have changed the word in the revised manuscript:
Line140-141:
- Figure 1, C: Please re-formulate “AAs number of MADS-box genes”. For example, with “the length of proteins potentially encoded by MADS-box genes”.
#RESPONSE: The comment is taken. We have changed the word in the revised manuscript:
Line142: (C) the length of proteins potentially encoded by MADS-box genes;
- Figure 2, legend: please italicize “Arabidopsis thaliana”.
#RESPONSE: Thank you for your careful reading. We have italicized the word:
Line167-170: (A) the phylogenetic tree of 126 MADS-box genes from cjaND genome and 101 MADS-box genes from Arabidopsis thaliana. The colored arc lines indicate the subgroups of MADS-box gene family; the colored letters indicate the canonical ABCE model genes.
- Line 111: “C. japonica” has to be italicized.
#RESPONSE: Thank you for your careful reading. We have italicized the word:
Line110-112: Currently, there are very few studies on the molecular regulation of ovule formation in C. japonica, and the understanding of seed abortion remains unclear.